# Opportunities and Challenges of Human IPSC Technology in Kidney Disease Research

**DOI:** 10.3390/biomedicines10123232

**Published:** 2022-12-12

**Authors:** Jia-Jung Lee, Chuang-Yu Lin, Hung-Chun Chen, Patrick C. H. Hsieh, Yi-Wen Chiu, Jer-Ming Chang

**Affiliations:** 1Division of Nephrology, Department of Internal Medicine, Kaohsiung Medical University Hospital, Kaohsiung Medical University, Kaohsiung 807, Taiwan; 2Faculty of Medicine, College of Medicine, Kaohsiung Medical University, Kaohsiung 807, Taiwan; 3Faculty of Renal Care, College of Medicine, Kaohsiung Medical University, Kaohsiung 807, Taiwan; 4Regenerative Medicine and Cell Therapy Research Center, Kaohsiung Medical University, Kaohsiung 807, Taiwan; 5Department of Biomedical Science and Environment Biology, College of Life Science, Kaohsiung Medical University, Kaohsiung 807, Taiwan; 6Human Disease iPSC Service Consortium, Institute of Biomedical Sciences, Academia Sinica, Taipei 115, Taiwan; 7Institute of Biomedical Sciences, Academia Sinica, Taipei 115, Taiwan

**Keywords:** pluripotent stem cells, human induced pluripotent stem cells, kidney disease, autosomal dominant kidney disease, SARS-CoV-2 infection

## Abstract

Human induced pluripotent stem cells (iPSCs), since their discovery in 2007, open a broad array of opportunities for research and potential therapeutic uses. The substantial progress in iPSC reprogramming, maintenance, differentiation, and characterization technologies since then has supported their applications from disease modeling and preclinical experimental platforms to the initiation of cell therapies. In this review, we started with a background introduction about stem cells and the discovery of iPSCs, examined the developing technologies in reprogramming and characterization, and provided the updated list of stem cell biobanks. We highlighted several important iPSC-based research including that on autosomal dominant kidney disease and SARS-CoV-2 kidney involvement and discussed challenges and future perspectives.

## 1. Introduction

Human pluripotent stem cells (PSCs) including embryonic stem cells (ESCs) are generated by in vitro-cultivated inner cell mass cells from the early conceptus, and induced pluripotent stem cells (iPSCs) can be derived from adult somatic cells [1,2]. Three important reasons for the application of human PSCs in clinical research are that they aid the study of human development, make the hard-to-obtain human cell types available for experiments, and can generate cells, tissues, and even organs for transplantation [3,4,5,6]. Stem cells are defined as having the capacity for both self-renewal and differentiation. Recalling the history, the German biologist Ernst Haeckel first brought up the term “stammzelle” (stem cell) in the scientific literature in 1868 with the fundamental questions of the continuity of the germ-plasm and the origin of the blood system [7]. The concept of repopulation from a single stem cell evolved with time and was confirmed nearly one hundred years later after definite evidence of common hematopoietic stem cells was provided by the work of James Till, Ernest McCulloch, and others in the 1960s [8]. Developmental and stem cell biologists further characterize the stemness based on the differentiation capacity of a single stem cell into totipotent, pluripotent, and multipotent stem cells. Totipotent cells in mammals indicate the first few blastomeres formed by the division of the zygote, which are able to become an individual. Most of the adult stem cells identified from different organ systems are multipotent cells with limited capacity for self-renewal. Multipotent cells can form a particular subset of cell types and cover most of the tissue-specific cell populations. On the other hand, PSCs including ESCs, as well as iPSCs, are able to propagate indefinitely, to produce all cell types, and to contribute to germ cells and are considered master cells [3,4,9,10].

## 2. The Evolving iPSC Technology: Generation, Characterization and Banking

Among PSCs, the iPSC technology bypasses the ethical issue related to the generation of ESCs and is preferred due to its eligibility for all ages of subjects without limitation theoretically. Human iPSCs were first generated by Professor Yamanaka’s group in 2007, one year after the group’s achievement of mouse iPSC generation [2,11]. To reverse a lineage-committed cell to a status with higher differentiation potency, the process of “reprogramming” requires the expression of four transcription factors, including OCT3/4, SOX2, KLF4, and c-MYC (OSKM, the Yamanaka factors) [2]. The stochastic model of reprogramming breaks down the molecular process into an early phase and a later phase. The early phase includes inhibition of the expression of somatic cell genes, mesenchymal-to-epithelial transition, and change in metabolism from oxidative phosphorylation to glycolysis. The later phase includes activation of pluripotency-associated genes, the suppression of tissue-specific transcription factors and developmental genes, and the bivalent methylation of H3K4me3 and H3K27me3 on high CpG promoter regions [5]. Over the years, multiple teams have successfully reprogrammed different types of somatic cells from various species, for example, pig, rabbit, monkey, goat, horse, cattle, chicken, and fish, which follow a common molecular mechanism. Many other sets of reprogramming factors have also been identified in addition to OSKM [5,12,13,14]. However, challenges still remain regarding the heterogeneity and inefficiencies of iPSC derivation, suggesting that other approaches may be required. The original method to reprogram human fibroblasts was to apply the retrovirus vector to introduce the exogenous OSKM. Other viral vectors such as lentivirus and adenovirus vectors showed promising and efficient results but raised the concern of genome integration. Our team applied the Sendai virus method for iPSC generation, which involves single-strand RNA replication in the cytoplasm without genome integration and is considered the safest of the viral methods [2,5,15,16,17,18].

More direct methods including DNA-based, RNA-based, and protein-based approaches have also been established. The initial approach with daily direct plasmid transduction could be modified by simultaneous expression of the EBNA-1 and the OriP sequences to enhance episomal amplification. The PiggyBac cassettes, one of the DNA-based vectors, work as a transposon and could be removed successfully from the host genome without a footprint. Recombinant B18R protein of the Vaccinia virus was used to minimize the host immune response toward the RNA plasmids. The protein-based method needs to fuse the protein with a cell-penetrating peptide to promote transduction. These methods are nonintegrating methods and are candidates for future cell therapies [5]. A comprehensive analysis was performed to compare several nonintegrating reprogramming methods including the Sendai viral (SeV), episomal (Epi), and mRNA transfection (mRNA) methods with the lentivirus method in human ESCs using a number of criteria, including the reprogramming efficiency, the success rate for generating at least three hiPSC colonies, the direct hands-on workload, the karyotype and genetic alternations, the time required for eliminating the exogenous reprogramming agents, and the RNA expressing patterns of undifferentiated and completely reprogrammed markers. The results indicated that all of the nonintegrating reprogramming methods generated high-quality human iPSCs; however, there were significant differences in the aneuploidy rate, gene delivery efficiency, success rate, and workload of the reprogramming process [19]. 

With the improved reprogramming methods, various human cell types have been successfully applied to generate iPSCs. These include fibroblasts, peripheral blood mononuclear cells (PBMCs), T cells, B cells, hepatocytes, cord blood, dental pulp, skin, cardiomyocytes, and other somatic cells [18,20]. Theoretically, all cells in the human body have the potential to be reprogrammed to iPSCs. With the advantages of being least invasive and easy to assess, PBMCs have become the most common cell source for human iPSC generation in biobanks [18,19,20,21,22,23,24]. 

Regarding the characterization of iPSCs, the most stringent criteria to demonstrate the pluripotency of mouse iPSCs are the tetraploid embryo complementation test and the ability to generate germline-competent adult chimeras. However, for human iPSCs, most chimeric experiment is withheld due to ethical concerns and strict regulation. In general, human iPSCs are assessed by the expression of the pluripotency genes, such as NANOG, OCT3/4, SOX2, etc. The differentiation capacity toward all the cells of the three germ layers can be confirmed by in vitro embryonic body formation and in vivo teratoma formation experiments [3,17,18,24]. For stem cell biobanks, most institutes include the following criteria for iPSCs: (1) ESC-like cell morphology; (2) silencing or removal of the transgenes; (3) expression of pluripotency and renewal markers; (4) confirmation of the differentiation potential into three germ layers; (5) karyotype analysis; (6) identity confirmation using DNA fingerprinting and short tandem repeat PCR with the parent cells; and (7) being free of biological contaminations by microbiological assay [18,21,22,24]. Other characterization methods, such as post-thaw viability, virus screening, and endotoxin, are important for cell manufacturing and clinical-grade application. More advanced analyses such as pluripotency analysis, copy number variation analysis, RNA-seq, exome seq, genotyping array, methylation array, mass spectrometry, whole-genome seq, and cellular phenotyping may be applied in different settings [21]. Comprehensive and stringent cell quality verification and validation are required for cell therapy trials. 

Several national or cross-institutional initiatives have established iPSC biobanks to generate, deposit, and deliver high-quality iPSCs for researchers or for therapeutic applications [18,20,22]. For example, the iPS Cell Research and Application (CiRA), Kyoto University, provides publicly available research-grade cell stocks including 29 human iPSC lines and 10 disease-specific iPSC lines. CiRA also works on establishing the iPSC stock with specified HLA and aims to provide clinical-grade allogenic iPSC cell stocks to achieve 50%, 80% and 90% coverage of Japan’s population [21]. In Europe, the European Bank for iPSC (EBiSC) supported by EUROPEAN Commission took a fast-track “Hot Start” process in 2014, which is renowned for establishing rigorous standardized pipelines and contributing to iPSC lines derived from 36 specific diseases. There are 852 donor-derived iPSC lines including 359 normal lines [23]. In the USA, government-funded biobanks such as the California Institute for Regenerative Medicine, the Coriell Institute for Medical Research, and the WiCell Research Institute and the privately owned Fujifilm Cellular Dynamics International are all active participants in iPSC research and applications [18,20]. The Taiwan Human Disease iPSC Service Consortium funded in 2015 by the government’s National Research Program and National Core Facility for Biopharmaceuticals was initially joined by five core facilities and ten partner hospitals. The consortium is now the only iPSC resource center in Taiwan and was transferred to be a core facility in Academic Sinica (National Academy of Taiwan). This core facility applies the Sendai virus reprogramming method and has banked 10 normal hiPSC lines and 74 disease lines with 23 individual disease types, which are all publicly accessible. The consortium also provides services and resources including hiPSC generation, characterization, iPSC-derived cells (including cardiomyocytes, retinal pigmented epithelial cells, neuronal progenitor cells, and cortical neurons), clustered regularly interspaced short palindromic repeats (CRISPR) transfection and single-cell generation iPSCs, and iPSC-related training courses to facilitate iPSC utilization and research development in Taiwan [17,18,20,25]. These advanced iPSC technologies and increasing resources have made human iPSC research more accessible and possible (Table 1).

## 3. iPSC-Based Human Disease Modeling: Kidney Differentiation and ADPKD

As iPSCs can be prepared from patient cells, it indicates that we may generate a library of patient somatic cells for recapitulating the key pathogenesis back to its early disease process. The human genetic background and unlimited cell sources make establishing iPSC-based disease models practical. Two advancing fields in iPSC-based disease modeling are neural disorders and cardiac diseases, both for which the cells needed are the most difficult to obtain [5]. The first patient-derived iPSC was from a case of amyotrophic lateral sclerosis (ALS) [5,26]. This motor neuron degeneration disorder is mostly sporadic and had no previous human models. Using the patient iPSC-derived motor neurons to model ALS disease progression, the researchers studied the cytosolic phenotype among the reported gene mutations. The cellular model recapitulated the cytosolic aggregation and verified some identified mutations on gene Tar DNA-binding protein-43 (TDP-43). Clonal variation and target cell variation were observed [5]. A novel approach using the iPSC-derived neuromuscular junction to model motor neuron diseases such as spinal muscular atrophy (SMA) was also reported [27]. For a prevalent neurodegenerative disease, Alzheimer’s disease, the iPSC-derived neurons from familial cases, sporadic cases, and healthy controls demonstrated the correlation between amyloid precursor protein (APP) proteolytic processing and P-tau. Moreover, the iPSC-based cellular model helped in discriminating the variations in treatment efficiency of docosahexaenoic acid (DHA) on cellular P-tau, which showed the promising role played by the patient-specific iPSC models for future drug screening. Cell therapy with human iPSC-derived dopaminergic progenitor cells for Parkinson’s disease is undergoing clinical trials [28]. Similarly, there are successful examples of using human iPSC-derived cardiomyocytes for modeling hypertrophic cardiomyopathy, dilated cardiomyopathy, and cardiac arrhythmias. Cellular phenotypes such as changing cell sizes, sarcomere disorganization, intracellular calcium homeostasis, prolonged ventricular repolarization, and increased risk of arrhythmia recapitulated the human disease process [5,29]. Large-scale drug screening platforms are in development, and cellular maturity is of note [30,31]. Preclinical and clinical trials for cardiac repair with PSCs including ESC- and iPSC-derived cells or cell sheets for treating heart failure or enhancing regeneration are in progress [32,33,34]. These in vitro disease models demonstrate the capability of iPSCs to reproduce the pathophysiological features and the genotyping–phenotyping systematic correlations.

To model human kidney diseases is of great challenge, since kidneys develop in three stages: pronephros, mesonephros, and metanephros. The metanephros is the final adult kidney, which requires a reciprocal interaction between two embryo tissues, the metanephric mesenchyme (MM) and the ureteric bud (UB). By deciphering and mimicking the natural developmental process, differentiation protocols start from the induction of the iPSCs to the endo–mesoderm to the critical stage of intermediate mesoderm (IM). Osafune et al. used the OSR1 gene reporter hiPSC line to optimize IM differentiation. Following sequential treatment with chemicals and growth factors, these committed kidney lineage cells were able to differentiate into adult renal cell types, including glomerular podocytes and renal tubule cells [35,36]. Taguchi et al. established a selective differentiation method through the posterior IM stage [37], and Takasato et al. reported the generation of kidney organoids containing glomeruli, renal tubules, collecting ducts, stromal cells, and vascular cells [38,39]. In 2015, the research groups of Freedman and of Morizane et al. published self-organized kidney organoids from hiPSCs showing delicate structures recapitulating the cellular repertoire and 3D structures [40,41,42]. Recently, Taguchi and Nishinakamura et al. reported a 3D method that directly differentiated iPSCs toward MM and UB to form high-order kidney organogenesis [43,44]. Osafune et al. used a different approach with 2D methods to generate MM and UB and aimed to increase the efficiency of cell generation, storage, and the reciprocal induction process and to enhance maturation and branching potentials [6,45,46]. 

The first iPSC-based kidney disease model was applied to autosomal dominant polycystic kidney disease (ADPKD), the most common monogenic kidney disease leading to progressive renal cysts and accounting for 5–10% of cases of kidney failure [47,48]. The causative genes of ADPKD are *PKD1* and *PKD2*, which encodes *Polycystin* 1 (PC1) and *Polycystin* 2 (PC2), respectively [49]. In 2013, Bonventre and Freedman et al. first applied iPSCs derived from ADPKD patients to model PKD. The PKD and control iPSCs exhibited comparable rates of cell proliferation, apoptosis, and ciliogenesis. However, as the iPSCs differentiated toward somatic epithelial cells, the patient iPSC lines showed decreased PC2 expression levels in the primary cilia, which suggested that PC1 regulated PC2 expression [50]. The same research group advanced the kidney differentiation protocol and applied the CRISPR and CRISPR-associated protein 9 (CRISPR-Cas9) technologies in modeling the ADPKD tissue-specific phenotype [41]. They reported that the human-PSC-derived kidney cells could self-organize into kidney organoids when the cells were treated with GSK3-beta inhibitor during the epiblast stage. The nephron-like structure contains proximal tubules, podocytes, and the endothelium both morphologically and functionally. Knockout of *PKD1* or *PKD2* induced cyst formation from kidney tubules in this model [41]. Little et al. supported the application of hiPSCs in modeling nephrogenesis and in identifying dysfunctional cellular pathways beyond the ciliopathic renal phenotype [38,51]. Morizane et al. identified the critical variable level of BMP4 in hiPSCs differentiating toward MM, which further differentiated toward renal vesicles and nephrons. The generated nephron organoids exhibited cell-type-specific responses to nephrotoxic agents [40]. After optimization, they reported a high efficiency of 80–90% production of nephron progenitor cells from hPSCs with a 9-day direct differentiation protocol [42]. On the other hand, a refined protocol was also in progress for UB differentiation, which showed a better cystogenesis phenotype of PKD [43,44,52]. The iPSC-derived UB showed PKD1 dose-dependent cyst formation in PKD modeling [44]. Researchers also tried to coculture MM and UB cells to enhance the functional maturity of iPSC-based organoids and disease models [45,46,52]. With mixed cell types, higher-order structures of nephrogenesis could be obtained [43]. These 3D cyst models or kidney cell on a chip and organ on a chip technologies are in development for future drug screening applications [53,54,55,56,57,58]. 

Our research team applied iPSC technology to decipher the direct or indirect effect of the mutation gene on the cardiovascular phenotype of ADPKD cases. We first generated three ADPKD patient-derived iPSC lines and confirmed their pluripotency as reported [17,59,60]. We then efficiently differentiated the iPSCs toward cardiomyocyte-like cells, which were characterized as ventricular cardiomyocytes. The electrophysiological analysis with calcium imaging, patch clamp, and drug challenges tests revealed the cell-specific characteristics and recapitulated the patient’s clinical phenotype [29]. iPSC-based experiments are invaluable in deciphering diseases with multiple organs involved. The isogenic control lines were generated and will be used in our future experiments [61] (Figure 1).

## 4. Challenges

In December 2019, a series of unknown-cause pneumonia cases emerged in Wuhan, Hubei, China. The causative pathogen was later identified as and named Severe Acute Respiratory Syndrome Coronavirus 2 (SARS-CoV-2). The infectious disease caused by this highly contagious novel pathogen was named coronavirus disease of 2019 (COVID-19) [62,63]. The world has been swept by the COVID-19 pandemic, which has resulted in more than 641 million infected cases and 6 million deaths so far. Scientists worldwide have worked relentlessly to tackle this challenge. The cellular entry of SARS-CoV-2 depends on the Angiotensin-Converting Enzyme 2 (ACE2) receptor which has species differences [64]. SARS-CoV-2 could not bind to mouse *ACE2* due to two amino acid differences at the critical virus-contacting residues. The multiple organ systems involved in COVID-19 patients also raised questions regarding whether it is a direct viral cytopathic effect or a consequence of systemic inflammation [65,66,67]. The human-iPSC-based experimental platform provided invaluable evidence for deciphering these tissue-specific effects. By means of the human intestinal organoids, direct viral infection and replication within the intestinal cells were identified [68]. SARS-CoV-2 also entered 3D human brain organoids, preferring targeted neurons, and was associated with altered distribution of Tau, hyperphosphorylation, and neuronal death. The results supported the potential neurotoxic effect of SARS-CoV-2 [69]. The iPSC-derived cardiomyocyte (iPSC-CM) experiments showed that SARS-CoV-2 can enter iPSC-CM via ACE2, replicate, and cause cytopathic effects including apoptosis and cessation of beating. Pathways involved including the activated innate immune response, antiviral clearance gene pathway, and inhibited metabolic pathway were identified [70]. An important article demonstrated that the platform developed using hPSC-derived cells and organoids could explore the SARS-CoV-2 viral tropism and cellular response to infection systematically. Their results concluded that cardiomyocytes, pancreatic endocrine cells, liver organoids, and dopaminergic neurons were permissive to the virus. These results obtained from hPSC-derived cells were validated with humanized mice, adult human islets, adult hepatocyte organoids, and adult cholangiocyte organoids [71]. For COVID-19 therapy exploration, direct infection of SARS-CoV-2 to human blood vessel organoids and kidney organoids engineered from ESCs and iPSCs could be inhibited by clinical-grade human recombinant soluble ACE2 [72]. A recent study observed the presence of SARS-CoV-2 nucleocapsid protein in the kidney proximal tubular epithelium and its association with tubular interstitial injury in COVID-19 patients. The group also investigated the direct impact of the virus on kidney cells using human iPSC-derived kidney organoids. Applying Little’s protocol, the generated kidney organoids contained cells expressing nephrin (the marker of podocyte), Lotus tetragonolobus lectin (the marker of the proximal tubule), ACE2 in the apical side, and E-cadherin (the marker of the distal tubule). Increased organoid fibrosis and increased collagen 1 protein expression improved after blocking TGFß, and inhibition of SARS-CoV-2 uptake by a protease inhibitor indicated that SARS-CoV-2 directly infects human kidney epithelial cells, including the stromal compartment. These results supported the direct cytopathic effect of SARS-CoV-2 in the kidney [73]. During this worldwide challenge of the COVID-19 pandemic, the PSC-based technology provides a human-cell-based, tissue-specific research platform, which is valid, invaluable, and can cover basic, preclinical, and clinically relevant experiments.

Clinically, we still have many unmet needs. Understanding patients’ pathological mechanisms is required for seeking effective treatment. By using the well-developed technique of next-generation sequencing, such as whole-exome sequencing or whole-genome sequencing, more and more disease-causing gene mutations can be identified. However, the phenotypes and molecular mechanisms caused by the gene mutations of the same disease cannot be well studied because of the lack of a suitable in vitro disease model to recapitulate the disease pathology and to connect the relationship between the phenotype and the genotype in the past. Nowadays, the genotype and phenotype of a disease can be correlated thanks to or with the invention of iPSCs. The 3D human kidney organoids derived from hPSCs combined with high-throughput screening (HTS) have established a powerful tool to enable toxicity screening, make disease phenotype recording in large volumes possible, and evaluate the safety and efficacy of future therapeutic regimens [74]. Furthermore, the use of isogenic control lines can pinpoint the molecular mechanisms associated with diseases, which may lead to the development of new therapeutic targets. The iPSC-dependent patient-specific disease models can significantly improve the vision for new pharmacological cures and drug screening [75]. Multiple 3D organoids can be further constructed to build a complex system, such as cardiovascular/respiratory systems, to mimic the physiological conditions and to provide controllable factors to study the disease mechanism in a more comprehensive way [76].

Regenerative medicine and cell therapy are still unmet needs in many fields. Renal transplantation and dialysis are the major options of current treatments for severe kidney damage. Shortage of donated organs is a major issue to evoke the development of tissue-engineered organs or parts of the kidney to meet clinical demand. The structural complexity and multiple functions of the kidney, including filtration, reabsorption, etc., make it challenging to rebuild a delicate kidney for clinical application. Based on Osafunes’ achievement, hPSCs can differentiate directly into kidney progenitors and Nephron Progenitor Cells (NPCs), including MM and UB. Not only were these progenitors able to form the nephron and collecting duct organoids but the *NPC*-derived glomeruli and renal tubules and UB-derived collecting ducts derived from the progenitors also interconnected with each other to form the functional construct of the kidney [6]. Various approaches including bioprinting to optimize differentiation for the high-content screen, scaffolding organoids on silk for scale-up of the structure, differentiating organoids on hydrogels to improve maturation, and culturing organoids within the *Polydimethylsiloxane* (*PDMS*) microchamber with flow to improve the developmental model were reported [74,77,78,79,80]. Although the combinatorial application of *NPC*s and *UB* with biomaterial scaffolds to construct a functional kidney is promising, it is still a long journey to create a de novo kidney for transplantation. To recover the function of the kidney with renal-cell-based treatment seems to be a practical remedy in the near future.

## 5. Conclusions

We are at a great time of the convergence of previous multiple disparate technologies, including iPSC generation and differentiation, mixed cell culture capabilities, next-generation sequencing and genome editing, 3D printing, sophisticated cell sensors, microfluidics, and microfabrication engineering. Integrating these advanced fields may lead us to the success of further genetic and phenotypic correlation and also to the physiological and functional maturation of an in vitro system. The substantial advances in gene editing enable isogenic control to serve as the best control in disease modeling and make the strategies for generating immune-compatible iPSC cell banks possible. The sophisticated cell sensors, image systems, and spatial transcriptomics demonstrated better organoid characterization, which makes great progress in establishing a drug screening system. Three-dimensional printing, microfluidics, and tissue engineering are efficient tools for solving vascular and size issues. The expanding pharmaceutical industry and rapid progression of space exploration also enhance the process of engineering tissue chip or organ-on-chip models [81,82,83,84]. Individualized disease models are considered critical tools for precision medicine. Organoid models derived from iPSCs have been used to reconstruct specific physiological and pathological conditions and environments. In precision medicine of many diseases, iPSC-based models are being evaluated in preclinical and clinical studies, for example, by retransplanting autologous epithelial organoids to rebuild an integrated intestinal barrier. Another particularly exciting feature of iPSC systems is the provision of insights into organ systems and disease-progressing processes, which cannot be monitored with clinical biopsies, such as the immune response in neurodegenerative disorders [85]. Precision medicine can offer patient-specific therapies for carriers of gene variants with undetermined significance [86,87]. These achievements can accelerate the research of orphan diseases and the development of precision medicine. Considering that the genetic background is specific and unique for every individual, this personalized approach can achieve the real vision of precision medicine. To well-translate the genotype to phenotype is urgently needed for both pathologic research and for time- and cost-efficient clinical trials. The future applications of human-PSC-based technology are on a broad spectrum and are promising to help humanity.

## Figures and Tables

**Figure 1 biomedicines-10-03232-f001:**
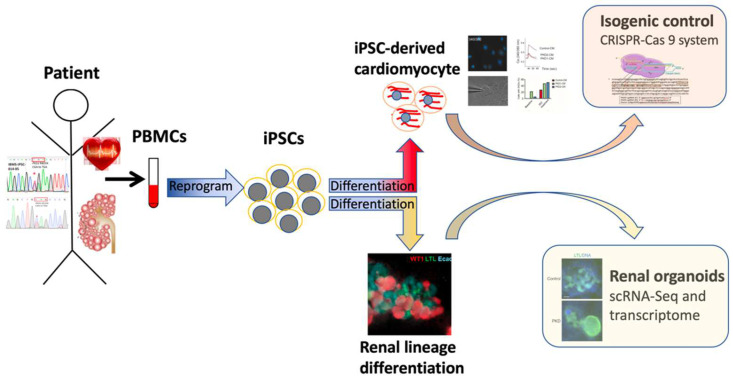
Schema of the induced pluripotent stem cell (iPSC)-based experimental approach for multiple-organ-involved diseases. Patients’ somatic cell-derived iPSCs can differentiate toward specific cell types to decipher multiple-organ-involved diseases, for example, autosomal dominant kidney disease. PBMC: peripheral blood mononuclear cell; iPSCs: induced pluripotent stem cells; CRISPR-Cas 9: clustered regularly interspaced short palindromic repeats and CRISPR-associated protein 9; scRNA-Seq: single-cell RNA sequencing.

**Table 1 biomedicines-10-03232-t001:** Brief information of iPSC repositories.

Name	Allies	Geographic Region	Products	Link
California Institute for Regenerative Medicine (CIRM)	Fujifilm Cellular Dynamics International (FCDI)	United States	40 diseases including 239 neurodevelopmental disorders, 131 liver disease, 442 heart disease, 65 neurodegenerative disease, 175 eyes disease, 191 lung disease, and 302 controls	https://www.cirm.ca.gov/researchers/ipsc-repository/about (accessed on 30 October 2022) https://www.fujifilmcdi.com/cirm-ipsc-products/ (accessed on 30 October 2022)
Center for iPS Cell Research and Application (CiRA)	ATCC, RIKEN, RUCDR	Japan	39 lines including 3 diseases: two neurodevelopmental diseases and a bone disorder	https://www.cira.kyoto-u.ac.jp/e/research/material_1.html (accessed on 30 October 2022)
European Bank for induced pluripotent Stem Cells (EBiSC)	HipSci	Europe	36 diseases, 895 iPSC lines including 359 normal control lines	https://ebisc.org/search (accessed on 6 December 2022)
Human Induced pluripotent Stem Cell Initiative (HipSci)	ECACC, EBiSC	United Kingdom	15 disease statuses, 339 disease lines, and 496 normal lines	https://www.hipsci.org/lines/#/lines (accessed on 30 October 2022)
Institute of Physical and Chemical Research (RIKEN)		Japan	14 disease categories including 231 diseases, 753 patients, and 3110 iPSC lines; 718 health control lines	https://cell.brc.riken.jp/en/hps/patient_specific_ips (accessed on 30 October 2022)
Human Disease iPSC Consortium Resource Center (Taiwan Human Disease iPSC Consortium)	BCRC	Taiwan	10 normal lines, 74 disease lines of 23 diseases	http://ipsc.ibms.sinica.edu.tw/schedule.html (accessed on 30 October 2022)
WiCell Research Institute (WiCell)	N/A	United States	1377 iPSC lines including 308 disease lines of 40 disease types	https://www.wicell.org/home/stem-cells/catalog-of-stem-cell-lines/advanced-search.cmsx (accessed on 30 October 2022)

## Data Availability

Not applicable.

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
