# Peer review of "Opportunities and Challenges of Human IPSC Technology in Kidney Disease Research"

_biomedicines, 2022, doi:10.3390/biomedicines10123232_

Round 1

Reviewer 1 Report

This manuscript requires extensive editing to remove multiple typos but also to make the story more coherent. I would suggest going back to the abstract and focusing on three components you have there - reprogramming, kidney and SARS-CoV2. Trim the unnecessary bits, i.e. remove sentences about first use of term stem cell, paragraphs about MND and cardiac modelling, trim the reprogramming section (table you provided was quite useful, I'd put it in the text where you discuss requirements for lines to be banked - if you mention it there, you'll save a lot of space that can be used to better describe iPSC-based kidney research and SARS-CoV2 which is really interesting. I feel like this may be a great paper but will need a fair bit of work to get there.

Reviewer 2 Report

In the manuscript entitled “Opportunities and challenges of human iPSC technology in

kidney disease researches”  Lee et al briefly overlook emerging field of cell reprogramming and differentiation. Particularly they address the issues of iPSCs differentiation into renal cell types and iPSC utility for Covid19 pandemic studies. Overall, the manuscript is interesting and introduces iPSC technology to general audience. English grammar ans style should be edited. There are also a number of remarks and recommendationas to the Authors. When recalling history of the term “stem cells” it would be appropriate to refer to the original papers. In the intro section Authors surprisingly avoided to explain pluripotency of the cells giving the examples of the toti and multipotent stem cells. Additionally I would like to note that zygote is not a stem cell because it is unable to self-renew. Also Authors have to reconsider their statement that PSCs “produce all cells types except the trophectoderm”. There are a number of studies that demonstrated that both esc and ipsc primed or naiive produce trophoectoderm DOI: 10.1126/sciadv.abf4416,  https://doi.org/10.7554/eLife.52504 and many other papers.

Round 2

Reviewer 1 Report

The paper reads better but can be further improved, please see my comments in the attached file.
